# Effect of the Rate of Nitrogen Application on Dry Matter Accumulation and Yield Formation of Densely Planted Maize

Juan Zhai, Guoqiang Zhang [ID], Yuanmeng Zhang, Wenqian Xu, Ruizhi Xie, Bo Ming [ID], Peng Hou, Keru Wang, Jun Xue *[ID] and Shaokun Li *[ID]

Institute of Crop Sciences, Chinese Academy of Agricultural Sciences/Key Laboratory of Crop Physiology and Ecology, Ministry of Agriculture and Rural Affairs, Beijing 100081, China
* Correspondence: xuejun@caas.cn (J.X.); lishaokun@caas.cn (S.L.)

**Abstract:** Planting maize (*Zea mays* L.) reasonably densely and adding amounts of appropriate nitrogen fertilizer are essential measures to improve the efficiency of maize yield and nitrogen use. In this study, two planting densities of $7.5 \times 10^4$ plants ha$^{-1}$ and $12.0 \times 10^4$ plants ha$^{-1}$ were established with the maize varieties DengHai 618 (DH618) and XianYu 335 (XY335). Simultaneously, 18 levels of nitrogen application were established, including a lack of nitrogen (N0) and increments of 45 kg ha$^{-1}$ nitrogen up to 765 (N765) kg ha$^{-1}$. The variables studied included the effects of the rate of nitrogen application on the characteristics of dry matter accumulation and the yield under drip irrigation, and they were integrated into water–fertilizer integration. The results indicated that the yield, harvest index, and dry matter accumulation of maize displayed a trend of increasing and then tending to be flat as the amount of nitrogen applied increased. The use of linear plus platform equation fitting indicated that the change in yield with nitrogen administered had the lowest turning point at N = 279 and N = 319, respectively. The next parameter that was measured was the harvest index. When highly dense maize was grown before silking, the rate of nitrogen applied was more obviously impacted by the accumulation of dry matter. The harvest index contributed 22.9–27.2% of the yield, and the total dry matter accumulation before and after silking contributed more than 70% of the production. Increasing the amount of nitrogen fertilizer is beneficial to prolonging the dry matter accumulation time and increasing the dry matter accumulation rate. The accumulation amount of dry matter was positively correlated with accumulation time and rate, and the correlation between dry matter and accumulation rate was greater. In conclusion, applying the right amount of nitrogen can dramatically increase the harvest index, accumulation of materials, and yield, with dry matter accumulation having the greatest influence on yield. The creation of dry matter is influenced by the time and rate of its accumulation, with its rate serving as the primary controlling factor.

**Keywords:** maize; dense planting; drip irrigation; water–fertilizer integration; material accumulation; harvest index



## 1. Introduction

Global food demand will continue to increase owing to population growth, and there may even be severe food shortages [1,2]. To meet the strict demand for grain caused by population growth, the output of grain may need to increase by 70% by 2050 [3,4]. The area of farmland is shrinking as urbanization dramatically increases, while the standard of living is also increasing. We can only meet the increasing demand for food by steadily increasing the amount of grain produced per unit area of currently used agricultural land. With 9% of the world's arable land, China has achieved a global milestone by feeding 22% of the world's population. The use of chemical fertilizer increases agricultural output and significantly contributes to global and Chinese food security [5,6]. However, the rapid expansion of the usage of chemical fertilizers poses environmental threats that impede the sustainable development of agriculture. A pressing concern in agricultural productivity is how to appropriately use nitrogen fertilizer to reduce nitrogen pollution.

Increasing density is an effective way to improve the grain yield of maize (*Zea mays* L.), and dense planting facilitates the accumulation of dry matter and the increase in maize yield [7]. The application of a reasonable amount of nitrogen and the timing of its application can increase dry matter accumulation after silking, meet the demand for high yield, and improve the efficiency of utilizing nitrogen fertilizer [8,9]. Biomass is not only the product of leaf sources, but also an essential source of yield formation. Higher dry matter accumulation is the basis of yield formation. Within a specific range, dry matter accumulation is directly proportional to the yield of maize grain [10–15]. Approximately 50–60% of the cumulative biomass of maize can be allocated to the grains [16,17]. Most of this biomass originates from the dry matter accumulation after flowering [8,18]. Therefore, increasing the dry matter after flowering is very important to increase grain yield.

The grain yield of crops is determined by the characteristics of dry matter accumulation, distribution, and transfer during the growth period, and there is a significant positive correlation between dry matter accumulation and yield [19]. Under the conditions of high-yield maize cultivation, many studies have been performed on material accumulation and nutrient absorption and utilization. It is clear that the accumulation and transportation of dry matter and nutrients in maize are primarily affected by different varieties of maize and cultivation measures. For example, for different maize varieties, Qi et al. [20] and Peng et al. [21] found that high-yielding or super-high-yielding maize varieties had higher rates of nutrient absorption and dry matter accumulation during their entire growth period. In particular, the ratio of nutrient absorption and distribution to grains after maize flowering was significantly higher than that of the low-yielding maize varieties. The increase in maize planting density can effectively improve the plant canopy structure, light interception capacity per unit area, and the dry matter production capacity of the population [10,22–24]. However, higher planting densities produce better results. Cao et al. [25] showed that a higher planting density results in a more efficient ability to transport nitrogen and reduces the imbalance of carbon and nitrogen metabolism in the plant and premature aging. The harvest index usually decreases with the increase in density [26,27]. The amount of dry matter accumulation after anthesis is crucial for obtaining a higher harvest index [28].

In 2020, the Crop Cultivation and Physiology Innovation team of the Chinese Academy of Agricultural Sciences created a record harvest of 24,948.75 kg ha$^{-1}$ in Qitai, Xinjiang [29]. The amount of nitrogen applied under different densities was optimized. The planting density was $7.5 \times 10^4$ plants ha$^{-1}$, and the recommended amount of nitrogen to apply was 340 kg ha$^{-1}$. When the planting density was $12.0 \times 10^4$ plants ha$^{-1}$, the recommended amount of nitrogen to apply is 380 kg ha$^{-1}$ [30]. Planting maize (*Zea mays* L.) reasonably densely and adding amounts of appropriate nitrogen fertilizer are essential measures to improve the efficiency of the maize yield and nitrogen use. The variables studied included the effects of the rate of nitrogen application on the characteristics of dry matter accumulation and the yield under drip irrigation, and they were integrated into water–fertilizer integration. However, it is not clear how the density and rate of nitrogen application affect dry matter accumulation and the yield of super-high-yield maize. Under the conditions of drip irrigation, water–fertilizer integration, and fractional fertilization, this study clarified the following: (1) how planting density and the rate of nitrogen application affect the dry matter accumulation characteristics of super-high-yield maize; and (2) how planting density and nitrogen application amount affect the yield of maize grain by affecting dry matter accumulation. The results will provide a theoretical basis for the scientific fertilization of super-high-yield maize.

## 2. Materials and Methods

### 2.1. Test Overview

The positioning was conducted from 2019 to 2021 at the Qitai Farm, Xinjiang (43°50′ N, 89°46′ E). The rainfall was 124–192.7 mm during the maize-growing season from April to October; the daily average temperature was 18.4 °C; the solar radiation was 3207.9–3577.8 MJ m$^{-2}$; the annual accumulated temperature that was ≥10 °C was 3160 to 3499.5 °C; and the

frost-free period was 156–181 days, respectively. The solar radiation, average temperature, and rainfall during the growing period are shown in Figure 1. The soil texture of the experimental field was sandy loam. The soil nutrient status before sowing was organic matter: 13.3 g kg$^{-1}$, N: 82.9 mg kg$^{-1}$, P: 53.8 mg kg$^{-1}$, K: 105.6 mg kg$^{-1}$, and a pH of 7.9.

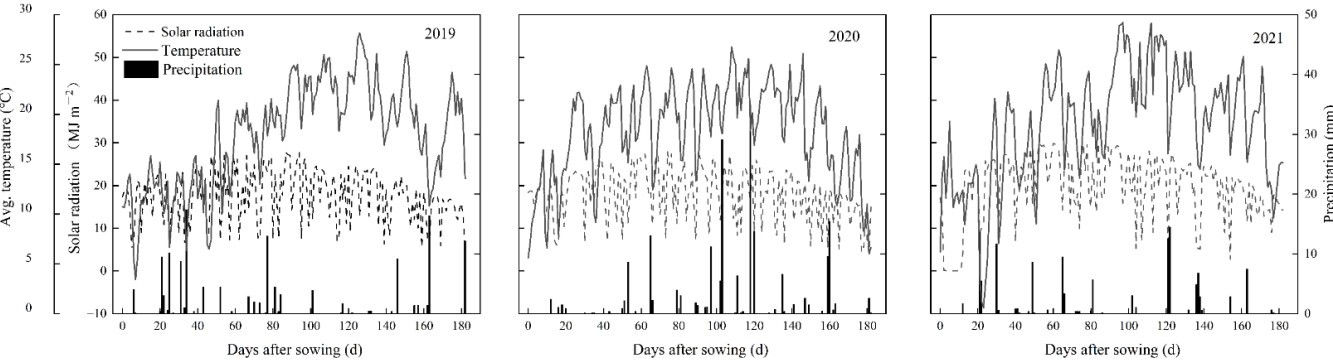

**Figure 1.** Solar radiation, average temperature, and rainfall during the maize-growing season from 2019 to 2021.

### 2.2. Test Design

In this experiment, two maize varieties that were tolerant to high densities were selected, including DengHai 618 (DH618) and XianYu 335 (XY335). Two planting densities were established, namely, 7.5 × 10$^4$ plants ha$^{-1}$ (farmer planting density) and 12.0 × 10$^4$ plants ha$^{-1}$ (high-yield planting density). A total of 18 levels of nitrogen were applied, including no nitrogen (N0), pure nitrogen 45 (N45), 90 (N90), 135 (N135), 180 (N180), 225 (N225), 270 (N270), 315 (N315), 360 (N360), 405 (N405), 450 (N450), 495 (N495), 540 (N540), 585 (N585), 630 (N630), 675 (N675), 720 (N720), and 765 (N765) kg ha$^{-1}$. The area of plot was 66 m$^2$ with three repetitions. Wide and narrow rows were used for planting, and the row spacing was 70 cm + 40 cm. The mode of irrigation and fertilization was an integrated drip irrigation system under film, water, and fertilizer. To accurately control the amount of fertilizer and water applied in each treatment, each community installed a set of 50 L differential pressure fertilizer tanks and fertilizer valves for fertilization.

### 2.3. Field Management

The planting date with a local temperature above 10 °C for 7 consecutive days was selected. The sowing date in 2019 was April 20, and the sowing date for both 2020 and 2021 was April 17. The harvest dates from 2019 to 2021 are October 4, October 2 and September 30, respectively. A total of 36 kg ha$^{-1}$ N, 108 kg ha$^{-1}$ P$_2$O$_5$, and 37.5 kg ha$^{-1}$ K$_2$O were applied as seed fertilizer before sowing, and N was not used during the entire growth period of N0. According to previous studies, under the condition of 12.0 × 10$^4$ plants ha$^{-1}$ density, the optimal irrigation amount is 5400 m$^3$, which can fully meet the water demand of maize during the whole growth period [31]. The other nitrogen application treatments were applied with water in equal proportions at the 9-leaf stage, 12-leaf stage, silking stage, 10 days after silking, and 20 days after silking. The seeds were watered on the first day after sowing to ensure uniform and rapid germination. To prevent late lodging and the hardening of seedlings, no irrigation was applied from sowing to 60 days after sowing. Chemical control (DA-6 Ethephon; China Agrotech, Shanxi, China) was applied at 600 mL ha$^{-1}$ during the V8–V10 period of maize growth. The pests, diseases, and weeds were well controlled.

### 2.4. Measurement Items and Methods
#### 2.4.1. Grain Yield

At physiological maturity, 20 ears were collected from the middle two rows in each plot, and the number of kernels on each ear was counted. After removing the border plots,

the final harvest was in a plot area of 10 m$^2$ with three replicates. The ear number, grain moisture content, and grain yield were also determined for each plot. The grain yield and kernel weight were expressed at a 14% moisture content.

### 2.4.2. Dry Matter Determination

At the 6-leaf stage (V6), 9-leaf stage (V9), 12-leaf stage (V12), silk stage (R1), milk ripening stage (R3), wax ripening stage (R5), and mature stage (R6) of the maize, three representative maize plants with consistent growth were randomly selected from each plot. The aboveground part of the plant was sampled from the base of the maize plant near the ground surface, and the stems, leaves, sheaths, male ears, female ears, bracts, and other organs were packed in sampling paper bags, marked for processing, and placed in an oven. The samples were incubated at 105 °C for 30 min to inactivate the enzymes and were then heated to a constant weight at 65 °C. The dry samples were weighed and recorded using a balance that was accurate to 0.01 g.

Dry matter accumulation amount after flowering = dry matter accumulation amount at the mature stage − dry matter accumulation amount at flowering stage  (1)

$$\text{Harvest index} = \text{grain yield} / \text{aboveground dry matter} \quad (2)$$

### 2.4.3. Maximum Rate of Accumulation and Amount of Dry Matter

Logistic curve fitting is one of the most commonly used models for dry matter in maize growth, with high accuracy. The dry matter accumulation at the different growth stages of the maize was fitted using the logistic curve [32]:

$$Y = W_{max} / (1 + ae^{-bt}) \quad (3)$$

where Y is the dry matter accumulation (t ha$^{-1}$); $W_{max}$ is the maximum accumulation (t ha$^{-1}$), and $t$ is the growth days (d). Parameters $a$ and $b$ are related to environmental conditions. The starting period ($t_1$), the ending period ($t_2$), the number of days between rapid accumulation (T), and the maximum accumulation rate ($V_{max}$) of the corresponding growth curve were obtained using the following equations:

$$t_1 = -\frac{1}{b}\ln\frac{2+\sqrt{3}}{a}; \; t_2 = -\frac{1}{b}\ln\frac{2-\sqrt{3}}{a} \quad (4)$$

$$V_{max} = (W_{max} \times b)/4; \text{T} = t_2 - t_1 \quad (5)$$

### 2.4.4. Calculation of the Rate of Contribution

The rate of contribution of the dry matter accumulation and harvest index to the yield were calculated using a correlation analysis [16]. The specific methods of calculation are shown as follows:

$$a_1 = \beta_1 Sx_1/S_y; b_1 = \beta_2 Sx_2/S_y; c_1 = \beta_3 Sx_3/S_y \quad (6)$$

$$C_{pre} (\%) = [a_1/(a_1 + b_1 + c_1)] \times 100\%; \quad (7)$$

$$C_{post} (\%) = [b_1/(a_1 + b_1 + c_1)] \times 100\%; \quad (8)$$

$$C_{HI} (\%) = [c_1/(a_1 + b_1 + c_1)] \times 100\% \quad (9)$$

In the formula, $a_1$, $b_1$, and $c_1$ are the standardized coefficients of pre-silking material accumulation, post-silking material accumulation, and harvest index, respectively. $\beta_1$, $\beta_2$, and $\beta_3$ are the coefficients of pre-silking material accumulation, post-silking material accumulation, and harvest index in a partial regression equation, respectively. $Sx_1$, $Sx_2$, $Sx_3$, and $S_y$ are the standard deviations of the pre-silking material accumulation, post-silking material accumulation, harvest index, and yield, respectively. $C_{pre}$, $C_{post}$, and $C_{HI}$ are the rates of contribution of the pre-silking material accumulation, post-silking material accumulation, and harvest index to grain yield, respectively.

### 2.5. Statistical Analysis

Microsoft Excel 2019 (Redmond, WA, USA) and SPSS 20.0 (IBM, Inc., Armonk, NY, USA) were used for the data analysis. A Pearson correlation analysis was used to analyze the correlation between dry matter, rate of dry matter accumulation, and accumulation time. Origin 2022 (OriginLab, Northampton, MA, USA) was used to complete all of the drawings.

## 3. Results

### 3.1. Effect of the Rate of Nitrogen Application on Maize Yield

With the increase of nitrogen application, the maize yield increased rapidly at the beginning, and the difference was not obvious when it reached a certain amount of nitrogen (Table 1). The results showed that when the rate of nitrogen application to the maize variety DH618 was 270–315 kg ha$^{-1}$ and 360 kg ha$^{-1}$ in 2019–2020, respectively, under a density of $7.5 \times 10^4$ plants ha$^{-1}$ and $12.0 \times 10^4$ plants ha$^{-1}$, the rate of nitrogen applied continued to increase. The yield did not increase, and the corresponding yields were 17.84–19.42 t ha$^{-1}$ and 18.53–20.5 t ha$^{-1}$, respectively. From 2019 to 2021, when the rate of nitrogen applied to the maize variety XY335 was 270–360 kg ha$^{-1}$ and 270–405 kg ha$^{-1}$ under a density of $7.5 \times 10^4$ plants ha$^{-1}$ and $12.0 \times 10^4$ plants ha$^{-1}$, respectively, the rate of nitrogen continued to increase, and the yield did not increase. The corresponding yields were 18.88–20.26 t ha$^{-1}$ and 19.32–21.68 t ha$^{-1}$, respectively. These results show that a higher yield can be obtained by increasing the planting density, but the demand for nitrogen fertilizer also increases. When the amount of nitrogen applied meets the growth demand of the maize, the yield will not be further improved if the amount of nitrogen applied is increased. At the same time, under the condition of sufficient water and fertilizer, the growth of maize was not inhibited.

**Table 1.** Effect of different nitrogen application rates on maize yield.

| Year | 2019 | | | | 2020 | | | | 2021 | |
|---|---|---|---|---|---|---|---|---|---|---|
| Density ($\times 10^4$ Plants ha$^{-1}$) | 7.5 | | 12.0 | | 7.5 | | 12.0 | | 7.5 | 12.0 |
| Varieties | DH618 | XY335 | DH618 | XY335 | DH618 | XY335 | DH618 | XY335 | XY335 | XY335 |
| Nitrogen Application (kg ha$^{-1}$) | t ha$^{-1}$ | | | | | | | | | |
| 0 | 13.02 g | 12.23 g | 12.12 h | 12.70 g | 9.97 f | 11.33 f | 8.76 h | 9.56 g | 8.42 j | 8.7 h |
| 45 | 13.28 g | 12.69 g | 12.66 h | 12.99 g | 10.10 f | 12.12 f | 9.23 h | 11.65 f | 9.58 i | 9.56 h |
| 90 | 13.81 f | 14.00 f | 13.55 g | 13.99 f | 12.84 e | 14.63 e | 10.99 g | 14.34 e | 11.08 h | 11.51 g |
| 135 | 14.81 e | 15.77 e | 14.33 f | 14.84 e | 14.91 d | 17.34 d | 13.03 f | 17.28 d | 12.62 g | 13.92 f |
| 180 | 15.49 d | 16.47 d | 14.88 e | 15.11 e | 16.72 c | 18.24 c | 14.91 e | 18.92 c | 14.58 f | 15.71 e |
| 225 | 16.87 c | 17.28 c | 15.85 d | 16.26 d | 18.16 b | 19.38 b | 16.42 d | 20.49 b | 16.05 e | 16.74 de |
| 270 | 17.26 bc | 18.13 b | 17.15 c | 16.86 c | 19.42 a | 20.26 ab | 17.76 c | 21.68 a | 16.85 de | 17.25 cd |
| 315 | 17.84 a | 18.88 a | 18.15 b | 18.40 b | 19.40 a | 20.70 a | 18.85 b | 22.25 a | 17.83 cd | 18.00 c |
| 360 | 17.62 ab | 18.69 ab | 18.53 ab | 19.32 a | 19.36 a | 20.49 a | 20.50 a | 22.65 a | 19.65 a | 19.24 b |
| 405 | 17.46 ab | 18.69 ab | 18.89 a | 19.10 a | 19.10 a | 20.68 a | 20.37 a | 21.85 a | 19.37 ab | 20.91 a |
| 450 | 17.42 ab | 18.42 ab | 18.75 ab | 19.17 a | 19.41 a | 20.14 ab | 20.48 a | 22.02 a | 19.34 ab | 20.53 a |
| 495 | 17.41 ab | 18.52 ab | 18.63 ab | 19.17 a | 19.11 a | 20.31 a | 20.19 a | 21.92 a | 18.96 abc | 20.27 ab |
| 540 | 17.61 ab | 18.64 ab | 18.83 a | 19.07 a | 18.89 ab | 20.25 ab | 20.48 a | 22.04 a | 18.97 abc | 20.86 a |
| 585 | 17.61 ab | 18.48 ab | 18.69 ab | 19.11 a | 18.96 a | 20.31 a | 20.22 a | 21.83 a | 18.54 abc | 20.78 a |
| 630 | 17.71 ab | 18.56 ab | 18.75 ab | 19.23 a | 18.87 ab | 19.85 ab | 20.42 a | 21.81 a | 19.43 ab | 20.56 a |
| 675 | 17.68 ab | 18.59 ab | 18.73 ab | 19.12 a | 18.88 ab | 20.13 ab | 20.19 a | 21.79 a | 19.17 ab | 20.41 ab |
| 720 | 17.80 ab | 18.39 ab | 18.67 ab | 19.11 a | 18.98 a | 19.85 ab | 20.22 a | 21.72 a | 18.66 abc | 20.35 ab |
| 765 | 17.56 ab | 18.58 ab | 18.68 ab | 19.11 a | 19.09 a | 19.98 ab | 20.31 a | 21.69 a | 18.25 bc | 19.99 ab |

Note: Values followed by different letters are significant at $p < 0.05$.

### 3.2. Effect of Nitrogen Application Rate on Maize Yield, Dry Matter Accumulation, and Harvest Index

The increase in nitrogen application, yield, harvest index, and dry matter accumulation before and after anthesis when the maize was grown at densities of $7.5 \times 10^4$ plants ha$^{-1}$ and $12.0 \times 10^4$ plants ha$^{-1}$, and the dry matter when the maize had matured, showed a trend of first increasing and then tending to be flat (Figure 2). A linear + platform was used to fit the rate of nitrogen application and yield, harvest index, and material accumulation

(Table 2). The results showed that under the two density conditions, the turning point of the change in yield with the rate of nitrogen applied was the lowest at N = 279 and N = 319, respectively. The second case in which the yield changed was the harvest index at N = 300 and N = 390, respectively. At a planting density of $7.5 \times 10^4$ plants ha$^{-1}$, there was a small difference in the accumulation of pre-silking and post-silking material in response to the application of N with values that ranged from N = 350 to N = 363, respectively. At a planting density of $12.0 \times 10^4$ plants ha$^{-1}$, there was a large difference in the response to N between pre-silking and post-silking material accumulation, and it ranged from N = 423 to N = 507, respectively. This showed that the maximum amount of dry matter was accumulated before silking. Compared with $7.5 \times 10^4$ plants ha$^{-1}$, the inflection points of N application for yield, harvest index, pre-flowering material accumulation, post-silking material accumulation, and dry matter accumulation at maturity increased by 14.3%, 30%, 44.9%, 16.5%, and 11.7%, respectively, at a density of $12.0 \times 10^4$ plants ha$^{-1}$. This indicates that the variation of population in the accumulation of dry matter in response to the application of N was more pronounced, particularly in pre-silking dry matter accumulation under high-density conditions, which implies that increasing the planting density requires more N fertilizer before conclusions can be drawn.

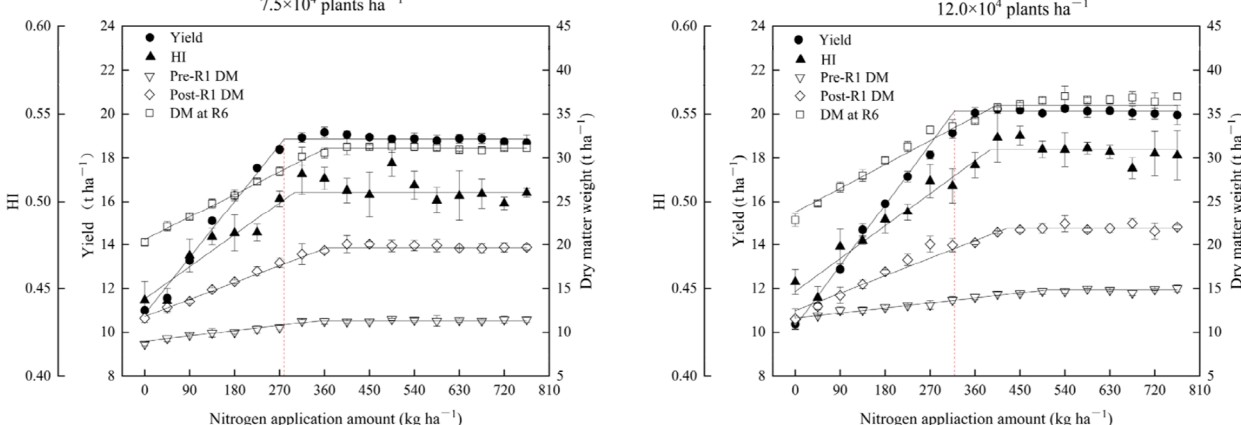

**Figure 2.** Effect of the nitrogen application rate on the yield, material accumulation, and harvest index of maize. The data are the average value of two varieties in three years under the same density and nitrogen application rate.

**Table 2.** Fitting equation between the nitrogen application rate and yield, material accumulation, and harvest index of maize.

| Plant Density ($\times 10^4$ plants ha$^{-1}$) | Indicators | Fitting Equation | $R^2$ |
|---|---|---|---|
| 7.5 | Yield | $Y = 10.82 + 0.03 \times N, N < 279; Y = 18.86, N \geq 279$ | 0.994 ** |
| | HI | $Y = 0.44 + 0.0002 \times N, N < 300; Y = 0.51, N \geq 300$ | 0.854 ** |
| | Pre-R1 DM | $Y = 8.98 + 0.007 \times N, N < 350; Y = 11.32, N \geq 350$ | 0.969 ** |
| | Post-R1 DM | $Y = 11.8 + 0.02 \times N, N < 363; Y = 19.61, N \geq 363$ | 0.997 ** |
| | DM at R6 | $Y = 20.58 + 0.03 \times N, N < 362; Y = 31.09, N \geq 362$ | 0.997 ** |
| 12.0 | Yield | $Y = 10.26 + 0.03 \times N, N < 319; Y = 20.14, N \geq 319$ | 0.994 ** |
| | HI | $Y = 0.45 + 0.0002 \times N, N < 390; Y = 0.53, N \geq 390$ | 0.973 ** |
| | Pre-R1 DM | $Y = 11.62 + 0.01 \times N, N < 507; Y = 14.85, N \geq 507$ | 0.993 ** |
| | Post-R1 DM | $Y = 12.46 + 0.02 \times N, N < 423; Y = 21.85, N \geq 423$ | 0.984 ** |
| | DM at R6 | $Y = 23.8 + 0.03 \times N, N < 402; Y = 35.97, N \geq 402$ | 0.977 ** |

Note: The data are the average value of two varieties in three years under the same density and rate of nitrogen applied. HI, harvest index; Pre-R1 DM, dry matter before silking; Post-R1 DM, dry matter after silking; DM at R6, dry matter at maturity. ** indicates a significant correlation at $p = 0.01$.

### 3.3. Rate of Contribution of Dry Matter and Harvest Index to Yield before and after Anthesis

The rate of contribution of the material accumulation and harvest index to yield was calculated before and after silking (Figure 3). The results showed that the dry matter accumulation after silking had the largest contribution to yield under both densities, and the rate of contribution of the harvest index to yield was 22.9~27.2%. The contribution of dry matter that accumulated before silking under low-density planting was higher than that under high-density planting. The dry matter accumulation before and after silking contributed more than 70% to the yield.

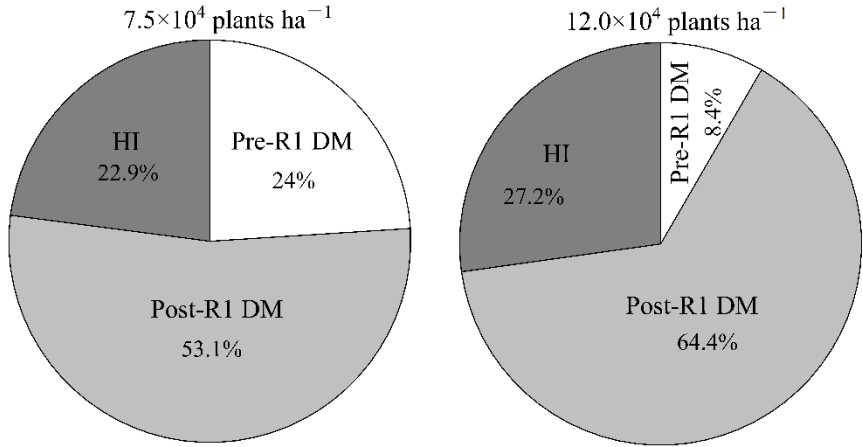

**Figure 3.** Rate of contribution of dry matter and the harvest index to yield before and after anthesis. Data are the average value of two varieties in three years under the same density and rate of nitrogen applied. HI, harvest index; Pre-R1 DM, dry matter before silking; Post-R1 DM, dry matter after silking; DM at R6, dry matter at maturity.

### 3.4. Effect of Nitrogen Application Rate on the Characteristics of Dry Matter Accumulation by Maize

Additional analysis showed that the dry matter accumulation of maize showed a trend of increasing with the number of days after sowing under different nitrogen application rates (Figure 4). With the progress in growth, the difference in dry matter accumulation between the different nitrogen rates gradually increased. Under the density of $7.5 \times 10^4$ plants ha$^{-1}$, the dry matter accumulation of the treatment with the highest yield increased by 38.5%, 13.6%, 34.1%, 18.5%, 23.8%, 31.5%, and 33.3% at the 6-leaf stage, 9-leaf stage, 12-leaf stage, silking stage, milk stage, wax stage, and mature stage, respectively, compared with the treatment without nitrogen application (N0). Under the density of $12.0 \times 10^4$ plants ha$^{-1}$, the dry matter accumulation of the treatment with the highest yield was 19.8%, 16.7%, 34.5%, 28.0%, 35.0%, 41.7%, and 42.2% higher than that of the treatment without nitrogen (N0) at the different growth stages, respectively.

The dry matter accumulation and days after emergence under the densities of $7.5 \times 10^4$ plants ha$^{-1}$ (Table 3) and $12.0 \times 10^4$ plants ha$^{-1}$ (Table 4) were fitted by a logistic equation. The fitting results, start time, end time, interval days, maximum accumulation rate, and maximum accumulation amount of dry matter can be calculated. The results showed that with the increase in the nitrogen application rate, the start and end times of the rapid accumulation of dry matter in the maize were delayed, and the end time of the rapid accumulation was delayed to a greater extent, which resulted in a longer duration of the rapid accumulation of dry matter. Simultaneously, with the increase in nitrogen accumulation, the rapid rate of accumulation of the dry matter increased. This shows that the nitrogen application rate can prolong the time of accumulation by delaying the termination period of rapid dry matter accumulation and also improve the rate of rapid dry matter accumulation, thus increasing the total dry matter accumulation.

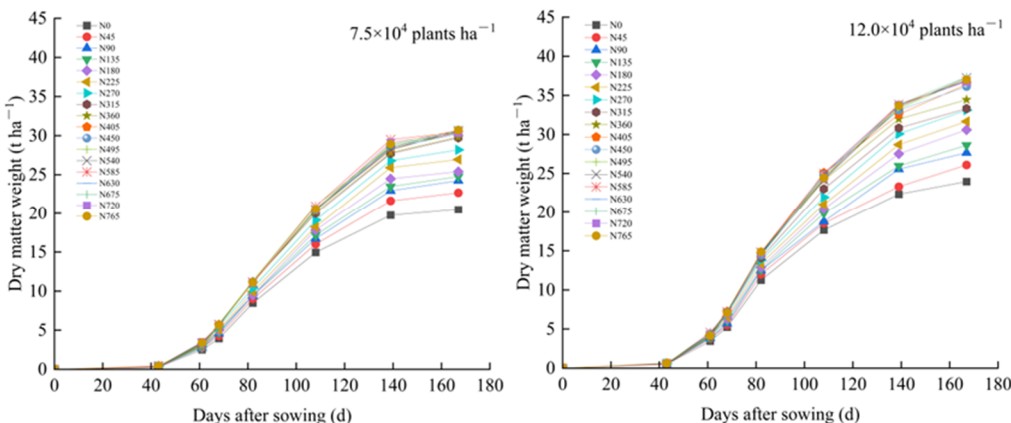

**Figure 4.** Effect of different nitrogen application rates on dry matter accumulation during the entire growth period of maize. The data are the average value of two varieties in three years under the same density and nitrogen application rate.

**Table 3.** Logistic fitting equation of density for $7.5 \times 10^4$ plants ha$^{-1}$.

| Nitrogen Application Amount (kg ha$^{-1}$) | Model | $R^2$ | Calculated Values | | | | |
|---|---|---|---|---|---|---|---|
| | | | $t_1$ (d) | $t_2$ (d) | $T$ (d) | $V_{max}$ (t ha$^{-1}$ d$^{-1}$) | $W_{max}$ (t ha$^{-1}$) |
| 0 | $Y = 20.52/(1 + 296.75e^{-0.06t})$ | 0.998 ** | 69.3 | 111.0 | 41.7 | 0.32 | 20.52 |
| 45 | $Y = 22.78/(1 + 254.88e^{-0.06t})$ | 0.998 ** | 69.9 | 113.5 | 43.6 | 0.34 | 22.78 |
| 90 | $Y = 24.39/(1 + 239.23e^{-0.06t})$ | 0.998 ** | 70.5 | 115.1 | 44.6 | 0.36 | 24.38 |
| 135 | $Y = 24.93/(1 + 238.15e^{-0.06t})$ | 0.999 ** | 70.4 | 115.1 | 44.6 | 0.37 | 24.93 |
| 180 | $Y = 25.72/(1 + 264.49e^{-0.06t})$ | 0.999 ** | 70.9 | 114.7 | 43.8 | 0.39 | 25.72 |
| 225 | $Y = 27.45/(1 + 237.47e^{-0.06t})$ | 0.998 ** | 71.6 | 117.0 | 45.4 | 0.40 | 27.45 |
| 270 | $Y = 28.54/(1 + 237.23e^{-0.06t})$ | 0.998 ** | 71.4 | 116.7 | 45.3 | 0.41 | 28.54 |
| 315 | $Y = 29.87/(1 + 222.30e^{-0.06t})$ | 0.998 ** | 71.1 | 116.9 | 45.8 | 0.43 | 29.87 |
| 360 | $Y = 29.94/(1 + 247.50e^{-0.06t})$ | 0.998 ** | 71.5 | 116.4 | 44.9 | 0.44 | 29.94 |
| 405 | $Y = 30.62/(1 + 229.53e^{-0.06t})$ | 0.999 ** | 71.6 | 117.4 | 45.8 | 0.44 | 30.62 |
| 450 | $Y = 30.95/(1 + 228.73e^{-0.06t})$ | 0.998 ** | 71.9 | 118.0 | 46.0 | 0.44 | 30.95 |
| 495 | $Y = 30.56/(1 + 255.84e^{-0.06t})$ | 0.999 ** | 71.1 | 115.4 | 44.3 | 0.45 | 30.56 |
| 540 | $Y = 30.72/(1 + 222.01e^{-0.06t})$ | 0.998 ** | 71.4 | 117.5 | 46.1 | 0.44 | 30.72 |
| 585 | $Y = 31.01/(1 + 264.38e^{-0.06t})$ | 0.999 ** | 71.7 | 116.0 | 44.3 | 0.46 | 31.01 |
| 630 | $Y = 30.54/(1 + 230.71e^{-0.06t})$ | 0.999 ** | 71.4 | 116.9 | 45.6 | 0.44 | 30.54 |
| 675 | $Y = 30.82/(1 + 227.23e^{-0.06t})$ | 0.999 ** | 71.5 | 117.4 | 45.9 | 0.44 | 30.82 |
| 720 | $Y = 30.89/(1 + 238.43e^{-0.06t})$ | 0.999 ** | 71.6 | 117.0 | 45.4 | 0.45 | 30.89 |
| 765 | $Y = 31.09/(1 + 229.13e^{-0.06t})$ | 0.999 ** | 71.8 | 117.7 | 45.9 | 0.45 | 31.09 |

Note: The data are the average value of two varieties in three years under the same density and nitrogen application rate. $t_1$, starting date of the rapid accumulation period; $t_2$, termination date of the rapid accumulation period; $T$, duration of rapid accumulation; $V_{max}$, maximal speed of accumulation; $W_{max}$, maximal accumulation. ** indicates a significant correlation between the $p = 0.01$ levels.

In different nitrogen application treatments, the time of dry matter accumulation and the maximum rate of accumulation increased first and then tended to be stable with the increase in amount of nitrogen (Figure 5). The dry matter accumulation time and the maximum rate of accumulation under the density of $12.0 \times 10^4$ plants ha$^{-1}$ increased to different extents by 4.43% and 11.74%, respectively, compared with the density of $7.5 \times 10^4$ plants ha$^{-1}$. The increase in dense planting and nitrogen application facilitates extension of the time of dry matter accumulation and an increase in the rate of dry matter accumulation.

**Table 4.** Logistic fitting equation of density for $12.0 \times 10^4$ plants ha$^{-1}$.

| Nitrogen Application Amount (kg ha$^{-1}$) | Model | $R^2$ | Calculated Values | | | | |
|---|---|---|---|---|---|---|---|
| | | | $t_1$ (d) | $t_2$ (d) | $T$ (d) | $V_{max}$ (t ha$^{-1}$ d$^{-1}$) | $W_{max}$ (t ha$^{-1}$) |
| 0 | $Y = 23.50/(1 + 228.70e^{-0.06t})$ | 0.997 ** | 65.8 | 107.9 | 42.1 | 0.37 | 23.50 |
| 45 | $Y = 25.30/(1 + 184.9e^{-0.06t})$ | 0.996 ** | 65.8 | 110.2 | 44.4 | 0.38 | 25.30 |
| 90 | $Y = 27.53/(1 + 152.02e^{-0.06t})$ | 0.995 ** | 66.9 | 114.4 | 47.5 | 0.38 | 27.53 |
| 135 | $Y = 28.22/(1 + 161.55e^{-0.06t})$ | 0.996 ** | 66.8 | 113.4 | 46.7 | 0.40 | 28.22 |
| 180 | $Y = 30.40/(1 + 141.93e^{-0.05t})$ | 0.996 ** | 67.8 | 116.8 | 49.0 | 0.41 | 30.40 |
| 225 | $Y = 31.58/(1 + 141.90e^{-0.05t})$ | 0.996 ** | 68.0 | 117.2 | 49.2 | 0.42 | 31.58 |
| 270 | $Y = 33.04/(1 + 150.13e^{-0.05t})$ | 0.996 ** | 68.4 | 117.2 | 48.7 | 0.45 | 33.04 |
| 315 | $Y = 33.11/(1 + 185.85e^{-0.06t})$ | 0.997 ** | 68.1 | 113.9 | 45.9 | 0.48 | 33.11 |
| 360 | $Y = 34.17/(1 + 208.97e^{-0.06t})$ | 0.997 ** | 68.3 | 113.0 | 44.7 | 0.50 | 34.17 |
| 405 | $Y = 35.80/(1 + 199.88e^{-0.06t})$ | 0.997 ** | 69.0 | 114.6 | 45.6 | 0.52 | 35.80 |
| 450 | $Y = 35.91/(1 + 198.72e^{-0.06t})$ | 0.998 ** | 69.2 | 115.0 | 45.8 | 0.52 | 35.91 |
| 495 | $Y = 36.82/(1 + 169.88e^{-0.05t})$ | 0.997 ** | 69.4 | 117.3 | 47.9 | 0.51 | 36.82 |
| 540 | $Y = 37.22/(1 + 158.728e^{-0.05t})$ | 0.997 ** | 69.9 | 118.9 | 49.1 | 0.50 | 37.22 |
| 585 | $Y = 36.73/(1 + 186.37e^{-0.06t})$ | 0.998 ** | 69.2 | 115.8 | 46.6 | 0.52 | 36.73 |
| 630 | $Y = 36.82/(1 + 184.78e^{-0.06t})$ | 0.998 ** | 69.7 | 116.7 | 47.0 | 0.52 | 36.82 |
| 675 | $Y = 37.50/(1 + 162.24e^{-0.05t})$ | 0.997 ** | 70.3 | 119.3 | 49.1 | 0.50 | 37.50 |
| 720 | $Y = 36.83/(1 + 179.79e^{-0.06t})$ | 0.997 ** | 70.0 | 117.5 | 47.6 | 0.51 | 36.83 |
| 765 | $Y = 36.97/(1 + 173.73e^{-0.06t})$ | 0.997 ** | 69.7 | 117.6 | 47.8 | 0.51 | 36.97 |

Note: The data are the average value of two varieties in three years under the same density and rate of nitrogen application. $t_1$, starting date of the rapid accumulation period; $t_2$, termination date of the rapid accumulation period; $T$, duration of rapid accumulation; $V_{max}$, maximal speed of accumulation; $W_{max}$, maximal accumulation. ** indicates a significant correlation between $p = 0.01$ levels.

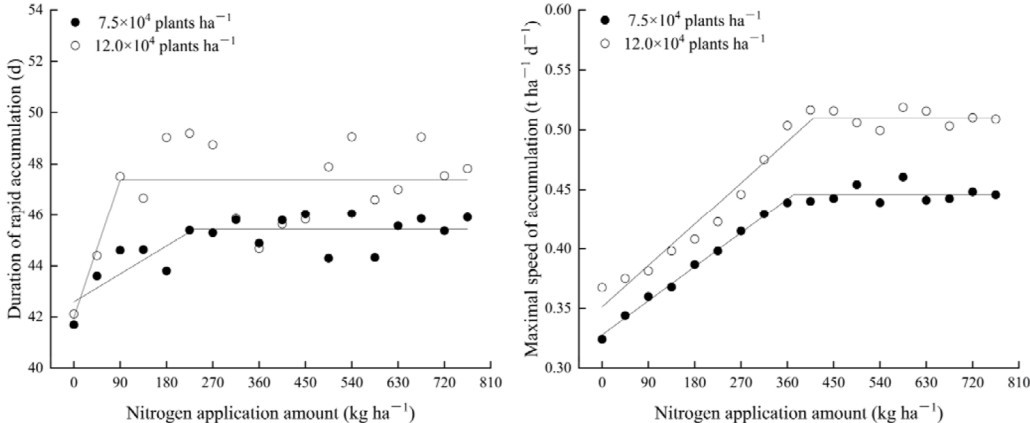

**Figure 5.** Effect of different nitrogen application rates on the duration and maximum accumulation rate of dry matter in maize. The data are the average value of two varieties in three years under the same density and nitrogen application rate.

### 3.5. Relationship between the Amount and Time of Dry Matter Accumulation and the Maximum Rate of Accumulation

As the yield gradually increased with the increase in nitrogen application, the dry matter at the mature stage was fitted with the time and rate of rapid accumulation of dry matter, respectively (Figure 6). The results showed that under the condition of $7.5 \times 10^4$ plants ha$^{-1}$, the duration and maximum rate of rapid dry matter accumulation significantly positively correlated with the dry matter accumulation, and there was a higher correlation between the dry matter accumulation and maximum rate of accumulation. Under the condition of $12.0 \times 10^4$ plants ha$^{-1}$, the maximum rate of dry matter accumulation significantly positively correlated with the amount of dry matter accumulated, while the correlation coefficient between the duration of rapid accumulation and the amount of dry matter accumulation did not reach a significant level. This shows that the rapid rate of accumulation of dry matter is the primary factor that affects the accumulation of dry matter.

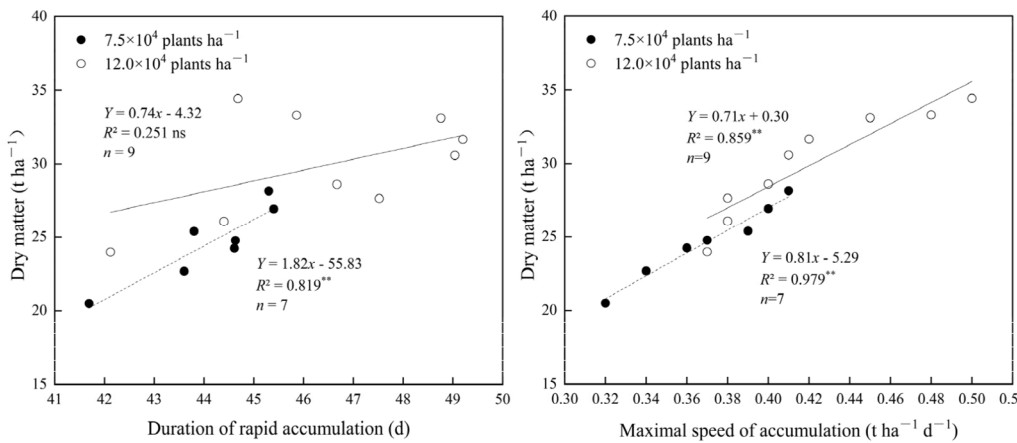

**Figure 6.** Relationship between the amount and time of dry matter accumulation and the maximum accumulation rate. The data are the average value of two varieties in three years under the same density and nitrogen application rates. ns, no significance. ** indicates a significant correlation between the $p = 0.01$ levels.

### 3.6. Path Analysis

A path analysis was conducted on the harvest index, rate of dry matter accumulation, accumulation time, and yield under different rates of nitrogen applications (Figure 7). The results showed that under the condition of $7.5 \times 10^4$ plants ha$^{-1}$, the amount of nitrogen applied primarily affected the yield by affecting the harvest index, duration of rapid dry matter accumulation, and the maximum rate. Under the condition of $12.0 \times 10^4$ plants ha$^{-1}$, the amount of nitrogen applied primarily affected the yield by affecting the harvest index and the maximum rate of dry matter accumulation.

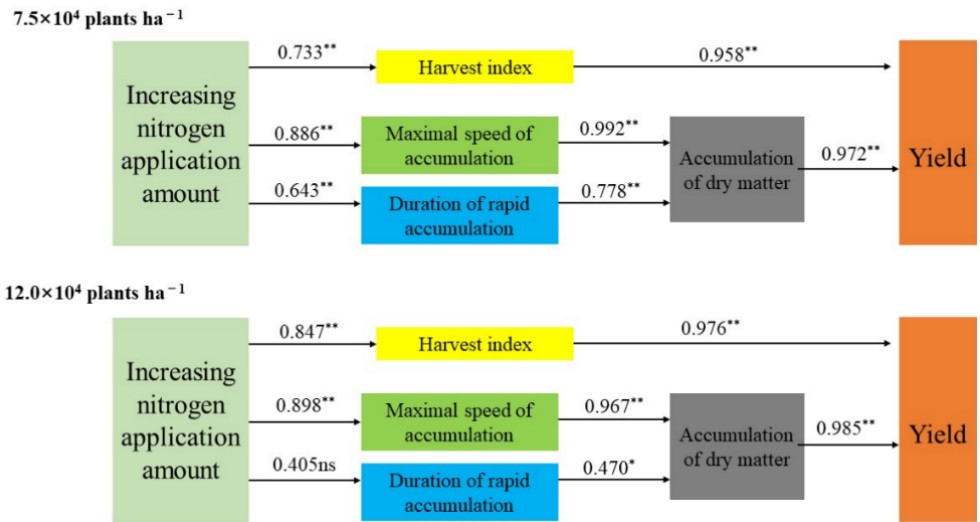

**Figure 7.** Path analysis of harvest index, dry matter accumulation, and yield. ns, no significance. * indicates a significant correlation between the $p = 0.05$ levels; ** indicates a significant correlation between the $p = 0.01$ levels.

## 4. Discussion

Dense planting and reasonable nitrogen application are the primary cultivation measures to increase maize yield. Increasing the planting density can increase the yield, but increasing the planting density also requires increasing the amount of nitrogen fertilizer to achieve a high yield [30,33]. Jin et al. [34] showed that when fertilizer was applied at different times for summer maize, the planting density of $7.5 \times 10^4$ plants ha$^{-1}$ was 184.5 kg ha$^{-1}$, and the yield reached its highest at 10.91 t ha$^{-1}$. The yield did not continue

to increase with the continued application of nitrogen fertilizer, but decreased owing to the influence of canopy light transmittance and the net photosynthetic rate. This study showed that with the increase in the nitrogen application rate, the maize yield showed a linear + platform trend. The highest yield of 19.42 t ha$^{-1}$ and 22.65 t ha$^{-1}$ was obtained under the density of $7.5 \times 10^4$ plants ha$^{-1}$ and $12.0 \times 10^4$ plants ha$^{-1}$ under 270 kg ha$^{-1}$ and 360 kg ha$^{-1}$ of applied nitrogen, respectively. When more nitrogen was applied, the yield did not increase, but there was no downward trend. This is because the research area is rich in light resources, which enables a higher planting density to obtain a high yield [35]. In addition, the super-high-yield level of 22.8 t ha$^{-1}$ maize was obtained under the density of $13.5 \times 10^4$ plants ha$^{-1}$ in this area [29]. Therefore, to obtain a higher yield and resource utilization efficiency, the planting density and amount of nitrogen applied should be reasonably allocated based on the local light radiation conditions in the high-yield maize cultivation system.

The amount of fertilizer can change the amounts of nutrients absorbed by crops at this stage, but it has little impact on the dynamic trend of dry matter and nutrient accumulation of the crops. Its growth shows an "S" curve change [36,37]. In this study, under the conditions of water and fertilizer integration and fractional fertilization, the change in dry matter in maize generally presents an "S" curve of a slow increase in the early stage, rapid accumulation in the middle stage, and slow growth in the late stage. The dry matter reaches its maximum before harvest, which is similar to the results of previous studies. However, some researchers have concluded that the application of nitrogen fertilizer can increase the dry matter accumulation of maize, but when the optimal nitrogen application amount is exceeded, there is a trend of decreasing the dry matter, which will result in a reduction in the crop yield [38–42]. The possible reason is that the increase in the amount of nitrogen applied can improve the leaf area index and dry matter of maize, but the photosynthetic effective radiation cannot be increased. This results in an inability of the lower leaves to conduct effective photosynthesis. In contrast, this study concluded that with the increase in nitrogen application, the dry matter increases first and then tends to stabilize, which will not cause a downward trend of dry matter. The primary reason is that the integrated water and fertilizer and fractional fertilization technology used in this experiment can effectively promote the absorption of nutrients in each growth stage of maize, maintain the green retention of leaves, delay leaf senescence, and prolong the photosynthetic time of leaves.

Some studies have shown that maize needs to absorb more nitrogen fertilizer after silking to meet the absorption needs of plants in the late growth stage [43,44]. Yang et al. [45] demonstrated that the excessive transportation of nutrients after the flowering of crops will affect the production of the photosynthetic products of the leaves in the later stage of crops, leading to accelerated leaf senescence, decreased grain filling rate, and limiting the increase in yield. However, nutrient transport that is too low does not facilitate grain filling, and it is difficult for maize to achieve high yields. Therefore, it is important to adjust the nutrient accumulation before and after the flowering of maize, maintain the coordination of source and sink, and increase the nutrient transfer amount and nutrient accumulation after flowering through appropriate fertilizer operation methods, which will play an important role in improving crop yields. The results showed that the suitable amount of nitrogen fertilizer facilitated the increase in matter accumulation before and after the anthesis of high-yield maize. Liu et al. [16] concluded that the increase in maize grain yield at a higher level of yield primarily depends on the increase in dry matter. This conclusion is similar to the results of this study. The amount of nitrogen primarily affects dry matter accumulation to improve yield. In the high-yield dense planting of maize, it is necessary to further improve the transport of dry matter to obtain a high yield.

The fundamental way to obtain a high yield is to improve the ability of plants to produce dry matter after flowering and the ability of dry matter to transport nutrients to the grains [12,46–49]. Ma et al. [40] showed that the rate of dry matter accumulation was the highest, and the yield was the highest when the nitrogen application rate was 306.5 kg ha$^{-1}$. Hou et al. [50] used the logistic model to simulate the dry matter accumulation of maize

under high-yield and ultra-high-yield cultivation modes and concluded that the maximum and average rates of dry matter increased significantly with the increase in the amount of nitrogen applied. This study showed that with the increase in the nitrogen application rate, the rate of dry matter accumulation showed a linear addition to the plateau. The primary reason could be that the nitrogen application rate extends the accumulation time by delaying the end period of rapid dry matter accumulation and increasing the rate of rapid dry matter accumulation, thereby increasing the dry matter accumulation.

## 5. Conclusions

With the increase in the nitrogen application rate, the maize yield, harvest index, and dry matter accumulation showed a linear + platform trend. Under the $7.5 \times 10^4$ plants ha$^{-1}$ density, the inflection point of the nitrogen application rate is 279 kg ha$^{-1}$, and the yield can reach 18.86 t ha$^{-1}$. Under the $12.0 \times 10^4$ plants ha$^{-1}$ density, the inflection point of the nitrogen application rate is 319 kg ha$^{-1}$, and the yield can reach 20.14 t ha$^{-1}$. The rate of contribution of the harvest index to yield was 22.9~27.2%, and the total rate of contribution of the dry matter accumulation to yield before and after silking was higher than 70%. Increasing the nitrogen application rate helps to prolong the dry matter accumulation time and increase the maximum rate of dry matter accumulation. The maximum rate of dry matter accumulation has a greater influence on the dry matter accumulation.

**Author Contributions:** Conceptualization, J.X. and S.L.; methodology, J.Z., J.X., G.Z. and Y.Z.; software, J.Z. and J.X.; validation, J.Z., J.X., G.Z., Y.Z. and S.L.; formal analysis, J.Z. and J.X.; investigation, J.Z., Y.Z. and W.X.; resources, S.L.; data curation, J.Z.; writing—original draft preparation, J.Z., J.X. and G.Z.; writing—review and editing, J.Z., J.X., G.Z., Y.Z., R.X., B.M., P.H., K.W. and S.L.; visualization, S.L.; supervision, J.X. and S.L.; project administration, J.X. and S.L.; funding acquisition, S.L. All authors have read and agreed to the published version of the manuscript.

**Funding:** This research was supported by grants from the Agricultural Science and Technology. Innovation Program (CAAS-ZDRW202004), China Agriculture Research System of MOF and MARA. Central Public Interest Scientific Institution Basal Research Fund (No. Y2022XK06).

**Institutional Review Board Statement:** Not applicable.

**Informed Consent Statement:** Not applicable.

**Data Availability Statement:** Not applicable.

**Conflicts of Interest:** The authors declare no conflict of interest.

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
