# Peer review of "Effect of the Rate of Nitrogen Application on Dry Matter Accumulation and Yield Formation of Densely Planted Maize"

_sustainability, doi:10.3390/su142214940_

Round 1

Reviewer 1 Report

This article is well written. The mechanism and law of the effect of nitrogen application rate on dry material accumulation and yield of densely planted maize are studied comprehensively. The experimental design part is detailed and reasonable, which is in line with the actual situation of maize planting and involves the actual problem of maize yield increase.In order to further improve the quality of the article, the following points need to be improved:

1. At the end of the introduction, it is suggested to add a paragraph, how is the research carried out in this paper and what is the purpose of the study.

2. In the Field management section, the basis for selecting the sowing date needs to be supplemented.

3. Figure 1 is not clear, especially Solar radiation and Temperature line.

4. In line160,the dry matter accumulation in different growth stages of maize was fitted using the Logistic curve, Please explain the reason. Whether other models are appropriate.

5. In line 197 and line 201, the nitrogen application rate continued to increase, and the reason why the yield did not increase.

6. In Section 3.3, the contribution of dry matter accumulation before silk laying under low density planting is greater than that under high density planting. Please explain the reason.

7. There are many complex factors to affect maize yield. Only explaining the amount of nitrogen applied is inappropriate, and it should be explained from a multi-factor perspective, such as the irrigation amount. As the planting density increases, whether the irrigation amount also increases? Whether the reason of increasing maize yield also due to the increase in irrigation amount.

Reviewer 2 Report

In the draft article, the authors examine the combined effect of nitrogen fertilization and seeding density of corn. The draft article is well developed and contains the relevant literature references. The material and method are appropriate, the results are traceable, and the conclusions are valid. I recommend the article for acceptance with some minor modifications.

My observations are as follows:

The draft article currently contains 41 references. It would be worthwhile to expand this a bit, especially with regard to the literature data published in the last 5 years.

108-110. the sentence in the line is not understandable, it is not necessary in the article.

The position of Figure 2 has shifted to the right, please change this.

Nice work, congratulations!

Reviewer 3 Report

This manuscript meets the requirements of Sustainability. However, the manuscript needs further improvement. Some suggestions and comments are as follows.

1. Pay attention to the type of paper.

2. Fit tests are required before path analysis.  

3. Conclusion section needs to be described in more detail.

4. The manuscript needs careful editing, e.g., Ma et al(line 394). There are other examples in the manuscript please check.

5. The references need to be updated with the latest research findings related to this manuscript

Round 2

Reviewer 1 Report

 It is recommended to accpet